# EVA-Flow: Learning Shared Chemistry for Unified Environment-Aware Molecular Conformation Generation

## Abstract

Molecular 3D structures underpin simulations of thermodynamics, kinetics, crystallization, and nucleation, making their accurate prediction central to drug discovery and materials design. However, molecular geometry can change dramatically across environments. Existing models either ignore environmental effects or train separate models for each environment. We hypothesize that transfer learning between different contexts is possible in conformer generation and that a general model will perform better in environments with limited training data: A very common and important setting including ligand binding and crystallization. We introduce *EVA-Flow*, a unified framework for environment-aware molecular conformation generation that learns shared chemical structure and adapts it through environment-conditioned generative modeling. Across four environments, vacuum, protein-ligand docking, solvation, and crystal packing, EVA-Flow substantially improves generation accuracy via pretraining and cross-environment finetuning. Analysis of molecules observed in multiple environments further shows that EVA-Flow produces distinct, physically valid, environment-specific conformations rather than memorizing a single canonical geometry.

## 1 Introduction

Molecules do not exist in a single fixed shape. A drug molecule may be synthesized in solution, crystallized for storage, redissolved in the body, and finally bind a target protein. At each stage the surrounding environment, whether solvent (Sobornova et al., 2024), crystal lattice (Chattopadhyay et al., 2025; Thompson & Day, 2014; Cruz-Cabeza & Bernstein, 2014), or protein pocket (Gao et al., 2010; Wang & Pang, 2007; Weikl & Paul, 2014), reshapes its 3D geometry.

Despite this, most generative models treat conformer generation as environment-agnostic: molecules are modeled in vacuum (Xu et al., 2022; Hassan et al., 2024) or within a single context such as protein–ligand docking (Corso et al., 2022; 2024), with a separate model for each setting. This fragmentation implicitly assumes that conformational distributions across environments are unrelated tasks. Yet while conformations change across environments, the underlying chemistry does not—the environment merely imposes physical forces that deform a shared molecular structure into different geometries.

This observation motivates a fundamentally different formulation: conformer generation as learning *shared invariant chemistry* under environment-dependent physical constraints, enabling a single model to transfer chemical knowledge across environments via environment-specific conditioning.

We propose *EVA-Flow*, a unified generative framework for environment-aware molecular conformation generation (Figure 1). EVA-Flow couples a variational autoencoder (VAE) (Kingma et al., 2019) encoder, which embeds both the molecular graph and its environment into a latent space capturing molecule–environment interactions, with a flow matching (FM) decoder (Lipman et al., 2022) that generates conformations conditioned on this representation.

Our contributions are:

- We formulate conformer generation as learning shared chemistry under environment-specific constraints, rather than training isolated models per environment.

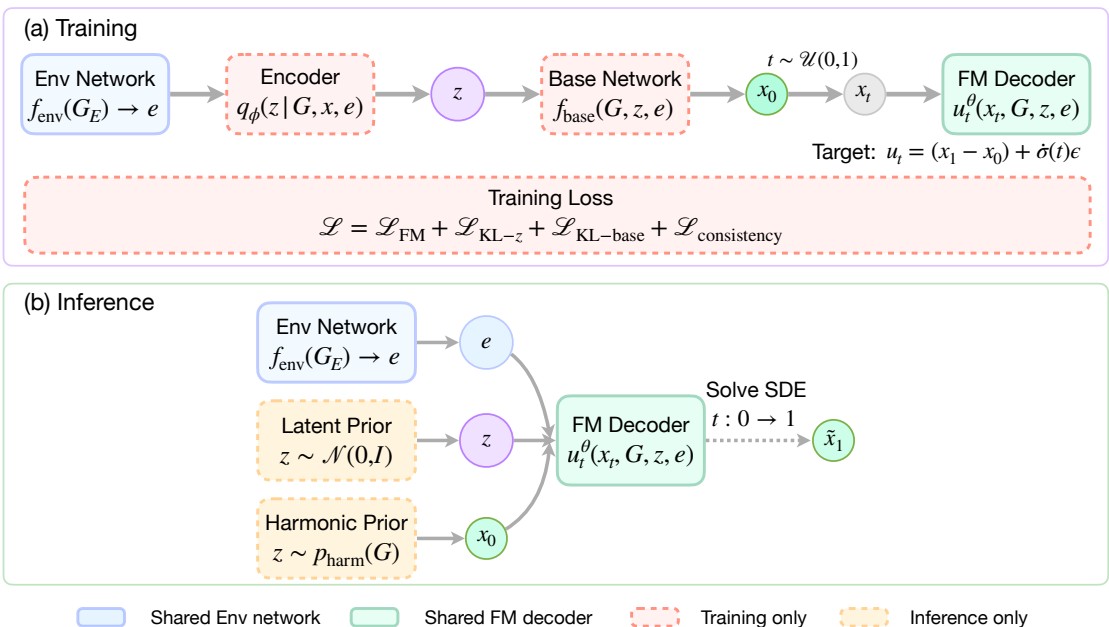

Figure 1: Architecture of EVA-Flow. **(a) Training.** The environment graph $G_E$ is embedded by $f_{\text{env}}$ into a vector $e$. The encoder $q_\phi(z \mid G, x, e)$ maps the molecular graph $G$, ground-truth positions $x$, and environment $e$ to a latent $z$. The base network $f_{\text{base}}(G, z, e)$ produces a distribution $p_0(x \mid G, z, e)$ from which $x_0$ is sampled. The FM decoder learns $u_t^\theta(x_t, G, z, e)$ with stochastic interpolation $x_t = (1-t)\,x_0 + t\,x_1 + \sigma(t)\epsilon$, targeting $u_t = (x_1 - x_0) + \dot{\sigma}(t)\epsilon$. Total loss: $\mathcal{L} = \mathcal{L}_{\text{FM}} + \mathcal{L}_{\text{KL-z}} + \mathcal{L}_{\text{KL-base}} + \mathcal{L}_{\text{consistency}}$. **(b) Inference.** The encoder and base network are bypassed: $e = f_{\text{env}}(G_E)$, $z \sim \mathcal{N}(0, I)$, and $x_0 \sim p_{\text{harm}}(G)$ are sampled independently. All three condition the FM decoder, which integrates $\dot{x}_t = u_t^\theta(x_t, G, z, e)$ from $t=0$ to $t=1$ with an SDE solver. The Env Network and FM Decoder share weights between training and inference.

- We introduce a unified VAE + flow matching architecture with an environment-conditioned encoder and decoder that handles diverse environments within a single model.

- We demonstrate that pretraining and cross-environment finetuning improve accuracy across four environments, and that the model produces distinct, physically valid conformations per environment rather than memorizing a single canonical geometry (verified by a UMA relaxation analysis, Appendix F).

## 2 Related Work

**Flow matching.** Flow matching (FM) trains a continuous normalizing flow by regressing the velocity field that transports a base distribution to the data, yielding a simulation-free objective (Lipman et al., 2022). Rectified flow (Liu et al., 2022) and conditional FM (Tong et al., 2023) extend this to straight-line paths and conditional generation. Equivariant (Hassan et al., 2024) and geometric (Chen & Lipman, 2023) variants adapt FM to 3D molecular settings.

**VAE + FM.** Hybrid VAE–flow models take two forms: (1) FM in the latent space replaces the VAE prior (Dao et al., 2023), or (2) FM serves as the decoder, mapping a base distribution to data conditioned on a latent code (Fischer et al., 2023; Sargent et al., 2025). We adopt the latter, using an environment-aware FM decoder for conformer generation.

**Molecular conformation generation.** Classical force-field and MD approaches are accurate but expensive. Neural methods generate conformers via equivariant diffusion (Xu et al., 2022), flows (Hassan et al., 2024), or torsion models (Jing et al., 2022). Most target a single environment—typically vacuum or

protein–ligand docking (Corso et al., 2024)—requiring separate models per setting. EVA-Flow addresses this limitation with a single model that conditions on structured environments.

**Diffusion in latent space for molecules.** Joshi et al. (2025) run diffusion in a learned latent space and decode to coordinates. EVA-Flow also uses a VAE encoder but performs FM in data space and explicitly conditions on the environment.

## 3 Environment-Aware Flow Matching

### 3.1 Problem Formulation

Let $G = (V, \mathcal{E})$ denote a molecular graph with nodes representing atoms and edges representing bonds. Each atom is associated with a 3D position $x \in \mathbb{R}^{3N}$, where $N$ is the number of atoms. A molecule exists in an environment $E$, such as a protein pocket, solvent box, or crystalline lattice. Our goal is to learn the conditional distribution: $p(x|G, E)$, capturing how environments modulate molecular conformations. Unlike prior works that train separate models for each $E$, we seek a unified model that generalizes across all $E$.

### 3.2 Model Architecture

Our framework, EVA-Flow, is a VAE with a FM decoder, consisting of multiple components (Figure 1).

1. **Environment Embedding Network**(Figure 1). The environment is represented as a graph $G_{\mathrm{E}} = (V_{\mathrm{E}}, \mathcal{E}_{\mathrm{E}})$, where nodes correspond to atoms with associated features and edges correspond to bonds or interactions. The construction of $G_{\mathrm{E}}$ is described in Section 4.1. The environment network, $f_{\mathrm{env}}$, consists of three graph convolutional network (GCN) layers (Zhang et al., 2019), followed by a two-layer MLP projection to the environment embedding vector $e \in \mathbb{R}^d$:

$$e = f_{\mathrm{env}}(G_{\mathrm{E}}) \tag{1}$$

   This embedding provides global information that conditions both the encoder and the decoder. Notably, GCN is not invariant or equivariant. We made this choice to keep the network consistent between environments $E$ both with and without explicit atomic positions in $\mathbb{R}^{N \times 3}$.

2. **Encoder** (Figure 1). The encoder is a GNN-based module that takes as input the molecular graph $G$, atomic positions $x$, and the environment embedding $e$. Input node features are formed by concatenating the atomic number, node attributes, and atomic positions. These features are processed by SE(3)-equivariant transformer (TorchMD-ET) (Thölke & Fabritiis, 2022; Hassan et al., 2024), producing node embeddings of size $[N, \text{hidden}]$. Each node embedding is then concatenated with the environment embedding $e$, and projected through two linear layers to produce the mean and log-variance for a node-level latent variational posterior distribution:

$$
\begin{aligned}
&q_\phi(z \mid G, x, e) \\
&= \prod_{i=1}^{N} \mathcal{N}\Big(z_i \mid \mu_{\phi,i}(G, x, e), \operatorname{diag}\big(\sigma_{\phi,i}^2(G, x, e)\big)\Big) \\
&= \mathcal{N}\big(\mu_\phi(G, x, e), \operatorname{diag}(\sigma_\phi^2(G, x, e))\big).
\end{aligned}
\tag{2}
$$

   where $\mu_{\phi,i}, \sigma_{\phi,i}^2 \in \mathbb{R}^d$ denote the mean and variance predicted for node $i$. $\mu_\phi, \sigma_\phi^2 \in \mathbb{R}^{N \times d}$ come from stacking all nodes into the appropriate vectorized format. Latent variable $z$ captures molecule–environment interactions.

3. **Base Distribution Network** (Figure 1). We parameterize a conditional Harmonic base distribution using an SE(3)-equivariant transformer (TorchMD-ET), which predicts the parameters of a Harmonic distribution over atomic coordinates. The base distribution is commonly parameterized with zero mean and the target molecule's graph Laplacian as precision matrix. We condition this distribution by predicting the mean and diagonal scaling:

$$\mu_{\mathrm{base}}, \lambda_{\mathrm{base}} = f_{\mathrm{base}}(G, x, e, z), \tag{3}$$

where $\mu_{\text{base}} \in \mathbb{R}^{N \times 3}$ is vector-valued and $\lambda_{\text{base}} \in \mathbb{R}^N$ is scalar-valued. Our conditional distribution is then:

$$L_{\text{base}} = L + \text{diag}(\lambda_{\text{base}}), \tag{4}$$

$$\Sigma_{\text{base}} = L_{\text{base}}^\dagger, \tag{5}$$

$$p_0(x_0 \mid G, x, e, z) = \mathcal{N}\Big(x_0 \,\Big|\, \mu_{\text{base}}, \Sigma_{\text{base}}\Big). \tag{6}$$

where $(\cdot)^\dagger$ is the Moore–Penrose pseudoinverse to handle global translational symmetries. Let $L_{\text{base}} = PDP^\top$ be the eigendecomposition, sampling is performed via:

$$x_0 = \mu_{\text{base}} + PD^{-1/2}\epsilon, \quad \epsilon \sim \mathcal{N}(0, I) \tag{7}$$

The network is trained by minimizing the KL divergence between the learned distribution and a Harmonic prior defined by the molecular graph topology $\mathcal{N}(\mathbf{0}, L^\dagger)$. During training the network accesses the ground truth $x$ for conditioning. The base network is not used at inference.

4. **FM Decoder** (Figure 1). The decoder generates conformations by transporting samples from the base distribution $p_0$ to the data distribution using stochastic flow matching. We adopt the TorchMD equivariant network to parameterize a time-dependent vector field $u_t^\theta$ that takes as input the current molecular structure $x_t$, the molecular graph $G$, the latent variables $z$, and the environment embedding $e$:

$$x_0 \sim p_0'(x_0) = \begin{cases} p_0(x_0 \mid G, x, z, e) & \text{Training} \\ \mathcal{N}\left(x_0 \mid \mathbf{0}, L^\dagger\right) & \text{Inference} \end{cases} \tag{8}$$

$$\dot{x}_t = u_t^\theta(x_t, G, z, e) \tag{9}$$

We use a stochastic interpolation between base and target samples. Given a base sample $x_0 \sim p_0'$ and a data sample $x_1 \sim p_{\text{data}}$, we define:

$$\begin{aligned} \mu_t &= (1-t)x_0 + tx_1, \\ x_t &= \mu_t + \sigma(t)\,\epsilon, \quad \epsilon \sim \mathcal{N}(0, I) \end{aligned} \tag{10}$$

We use the noise schedule:

$$\sigma(t) = \sigma\sqrt{t(1-t)} \tag{11}$$

5. **Inference** (Figure 1). At inference time, given the environment graph $G_E$, we compute the embedding $e = f_{\text{env}}(G_E)$ and sample a latent $z \sim p(z) = \mathcal{N}(0, I)$. The initial positions are sampled from the simple Harmonic prior $x_0 \sim \mathcal{N}(\mathbf{0}, L^\dagger)$. Starting at $x_{t=0} = x_0$, we sample using an SDE discretization with additive Gaussian noise. For a decreasing time schedule $\{t_i\}_{i=0}^n$ (from $t_0$ to $t_n$), we perform Euler–Maruyama updates

$$\begin{aligned} x_{t_{i-1}} &= x_{t_i} + u_\theta(x_{t_{i-1}}, t_{i-1}; G, z, e)\,\Delta t_i \\ &\quad + \sqrt{\sigma^2(t_i) - \sigma^2(t_{i-1})}\,\xi_i, \\ \xi_i &\sim \mathcal{N}(0, I), \qquad \text{with COM}(\xi_i) = 0, \end{aligned} \tag{12}$$

where $\sigma(t)$ is a linear noise schedule (implemented as $\sigma^2(t) = t^2$) and $\Delta t_i = t_{i-1} - t_i$. The final conformer $\hat{x}$ is obtained after $n$ steps.

## 3.3 Training Objectives

We jointly optimize the encoder, environment embedding network, base distribution network, and FM decoder using a composite objective. The total loss consists of four terms, each serving a distinct role (Figure 1):

1. **Latent Prior Regularization**. This is the typical VAE loss to regularize a latent variable $z$ towards a normal:

$$\mathcal{L}_{\text{KL}-z} = D_{\text{KL}}(q_\phi(z|x, G, e) \,||\, \mathcal{N}(0, I)). \tag{13}$$

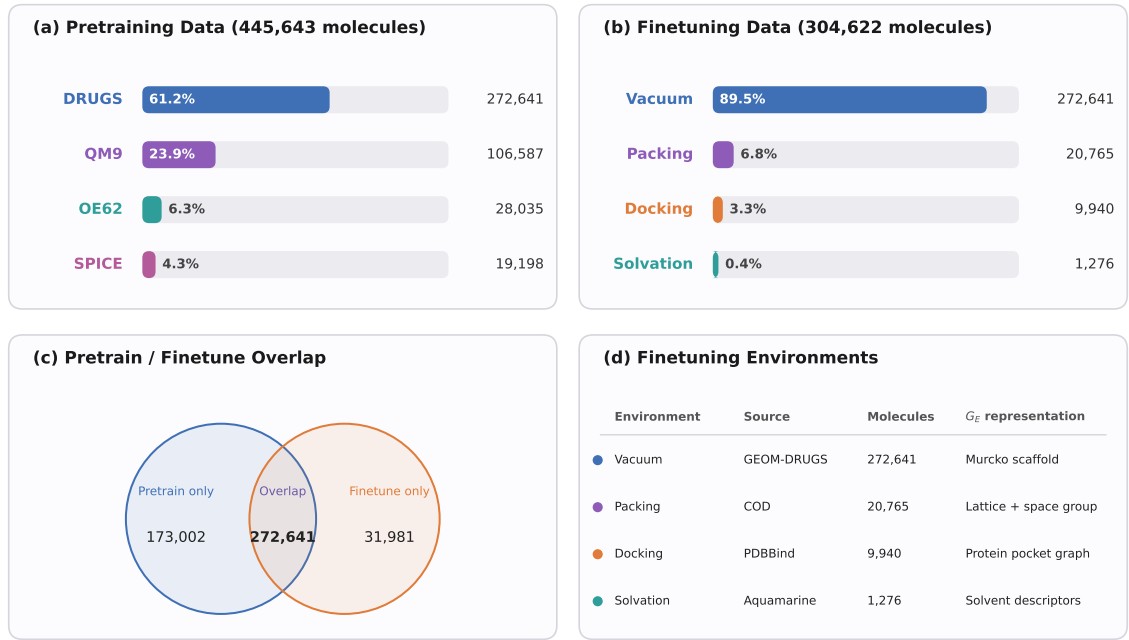

Figure 2: Dataset composition. **(a)** Pretraining corpus by source (445,643 molecules). **(b)** Finetuning corpus by environment (304,622 molecules). **(c)** Overlap between the pretraining and finetuning sets: all 272,641 GEOM-DRUGS molecules used for Vacuum finetuning are also part of the pretraining corpus. **(d)** Data source and environment graph $G_E$ representation for each finetuning environment.

2. **Base Distribution Regularization**. The learned base distribution is regularized by KL divergence toward a simple Harmonic prior induced by graph topology $L$:

$$\mathcal{L}_{\text{base}} = D_{\text{KL}}\big(\mathcal{N}(\mu_{\text{base}}, \Sigma_{\text{base}}) \,\|\, \mathcal{N}(0, L^{\dagger})\big). \tag{14}$$

3. **Latent Consistency Loss**. This loss encourages the latent representation $z$ to capture stable molecule–environment interactions. We penalize disagreement between the encoder's posterior mean given target conformation $x$ and initial conformation $x_0 \sim p_0'(x_0)$:

$$\mathcal{L}_{\text{consistency}} = ||\mu_{\phi}(x_0, G, e) - \mu_{\phi}(x, G, e)||_2^2. \tag{15}$$

4. **FM Loss**. This loss trains the decoder to transport samples from the base distribution to the data distribution through continuous-time dynamics.

$$\mathcal{L}_{\text{FM}} = \mathbb{E}_{t, x_t}[||u_t^{\theta}(x_t, G, z, e) - u_t(x_0, x)||^2], \tag{16}$$

where $u_t$ is the analytically defined target velocity field:

$$u_t = (x_1 - x_0) + \dot{\sigma}(t)\,\epsilon, \quad \epsilon \sim \mathcal{N}(0, I),$$
$$\dot{\sigma}(t) = \sigma \frac{1 - 2t}{2\sqrt{t(1 - t)}} \tag{17}$$

All networks are trained jointly at each iteration. We chose to use uniform weighing of the various loss terms:

$$\mathcal{L} = \mathcal{L}_{\text{KL}-z} + \mathcal{L}_{\text{KL}-\text{base}} + \mathcal{L}_{\text{consistency}} + \mathcal{L}_{\text{FM}}. \tag{18}$$

## 4 Experiments

### 4.1 Datasets

**Pretraining**. For large-scale pretraining, we combine datasets containing conformations of small organic and drug-like molecules. This includes relaxed conformations from GEOM-QM9 and GEOM-DRUGS (Axelrod & Gomez-Bombarelli, 2022; Nikitin et al., 2025), as well as higher-energy conformations from the SPICE PubChem subset (Eastman et al., 2023). We further performed relaxation on SPICE dataset using `CREST` (Pracht et al., 2024) to obtain low-energy conformations. In addition, we include the OE62 dataset (Stuke et al., 2020), which consists of crystal-forming molecules relaxed in vacuum. Altogether, the pretraining corpus contains ∼446K molecules and 9.8M conformations (Figure 2a).

We construct the environment graph ($G_E$) during pretraining. For each molecule, we apply the RDKit (Landrum & contributors, 2025) `MurckoScaffold` package (RDK, 2025) to extract the molecule's Bemis–Murcko scaffold (Bemis & Murcko, 1996) by removing substituents and retaining only the core ring systems and the linkers connecting them. The partial graph is passed to the environment network (Figure 2a).

**Finetuning**. We finetune our model in four environments:

1. **Vacuum**. Molecules in vacuum are isolated. We use GEOM-DRUGS dataset. We build the environment graph similar to pretraining (Figure 2(d)).

2. **Protein–Ligand Docking**. In this task, molecules (ligands) adapt to protein binding pockets. We use PDBBind-v2020 dataset (Liu et al., 2015). The environment graph is constructed using heavy atoms in the protein pocket. We add edges between atoms in the same protein residue. The atomic positions of the binding pocket are included (Figure 2(d)).

3. **Solvation**. Dissolved molecules interact with the solvent. We use the Aquamarine dataset (Medrano Sandonas et al., 2024), which includes conformations relaxed in implicit water. To construct the environment graph, we use a water molecule and add solvation descriptors to the node features such as cavity surface area (sCAV), cavity volume (vCAV), free energy in electrolyte (eSOLV), and non-electrostatic free energy (eNELEC)(Figure 2(d)). These descriptors represent the interaction between molecules and solvent.

4. **Crystal Packing**. Molecules interact with neighboring molecules in crystals. We use the Crystallography Open Database (COD) (Gražulis et al., 2009; 2012), and filter to the small organic molecular crystals. We convert the fractional coordinates from the CIF files to Cartesian positions, and recover the bonds with `OpenBabel` (O'Boyle et al., 2008; 2011). For the environment graph, we extract the partial graph similar to pretraining. We also add to the node feature the space group number (sg) and lattice parameters $(a, b, c, \alpha, \beta, \gamma)$ (Figure 2(d)).

Further details about data processing and the distributions of number of atoms and the number of rotatable bonds can be found in Sections A and B.

### 4.2 Experimental Setup

We compare four training strategies.

1. **Pretraining + Individual Finetuning**. We first pretrain the model on the pretraining datasets, then finetune it on a single environment. In the pretraining, the model is exposed to massive molecules and conformations to learn shared chemistry. In the individual finetuning, the model learns a specific environment.

2. **Pretraining + Unified Finetuning**. We first perform training on the pretraining datasets (Section 4.1), but then finetune on all environments of the finetuning datasets. This setup examines if model learns to adapt to different environments.

3. **Unified Finetuning**. we skip the pretraining and only train the model across all environments.

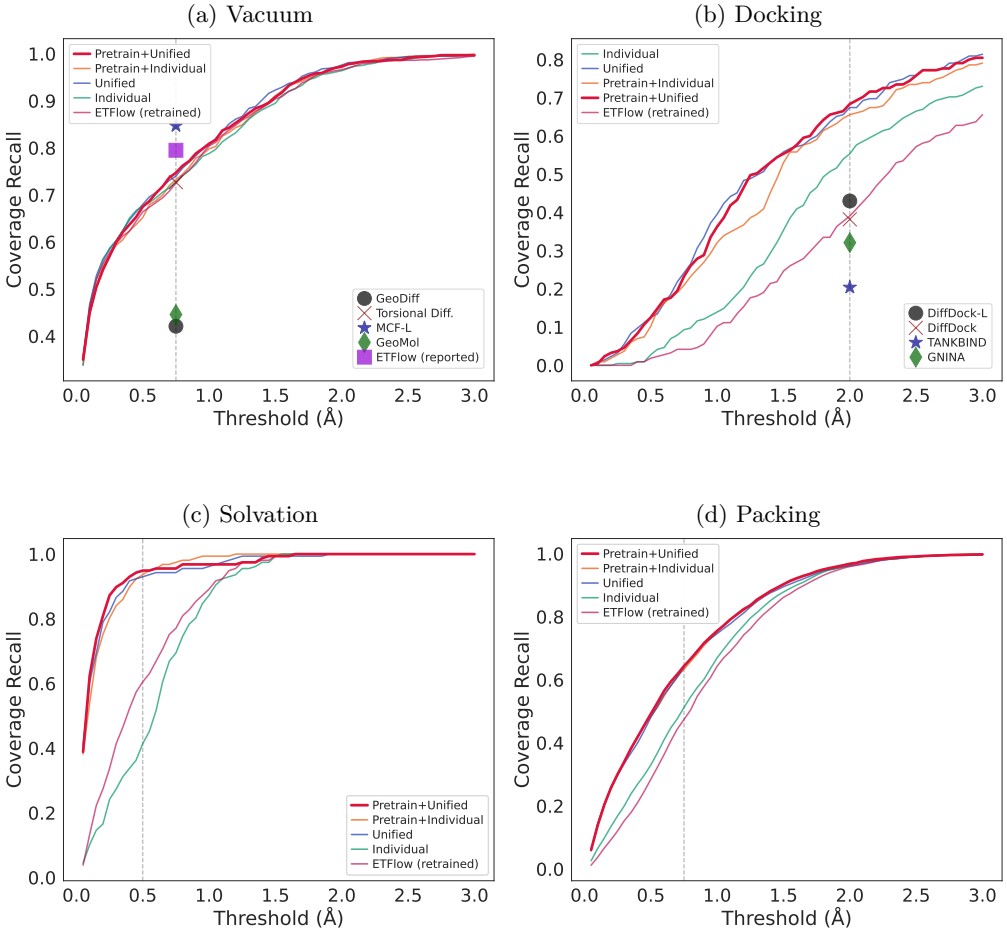

Figure 3: Recall COV as a function of the RMSD threshold across the four environments. In every panel the four EVA-Flow training setups (Individual, Unified, Pretrain+Individual, Pretrain+Unified) are compared to retrained ETFlow. **(a)** Vacuum (cleaned GEOM-DRUGS (Nikitin et al., 2025)); reference points denote published Recall COV from Xu et al. (2022); Ganea et al. (2021); Jing et al. (2022); Wang et al. (2024); Hassan et al. (2024). **(b)** Docking (PDBBind); reference points denote published Recall COV from Lu et al. (2022); McNutt et al. (2021); Corso et al. (2022; 2024). **(c)** Solvation (Aquamarine) and **(d)** Packing (COD); these environments have no established conformer-generation baselines, so we compare against retrained ETFlow only. Dashed vertical lines mark the environment-specific RMSD thresholds (0.75/2.0/0.5/0.75 Å).

4. **Individual Finetuning**. We train separate models for each environment, without pretraining. This setup is a baseline to study the effects of unification and pretrain.

**Evaluation protocols**. We use two protocols. The **4-conformer protocol** is our default evaluation across all four environments: for every test molecule we take its single ground-state reference conformer and ask each model to generate 4 conformers (Table 1, Figure 3). It is uniform across environments, and reflects the practical setting where only one reference geometry is available. Because each molecule contributes one reference and four generations, per-molecule COV can only take values in $\{0, 1\}$ for Recall and $\{0, 0.25, 0.5, 0.75, 1\}$ for Precision; we therefore report mean COV only and supplement it with the full mean / median AMR. The **2K protocol** is used only for the Vacuum benchmark, to allow direct comparison with published baselines (Ganea et al., 2021; Jing et al., 2022; Wang et al., 2024; Hassan et al., 2024). Under this protocol, each molecule with $K$ ground-truth conformers leads to $2K$ generated conformers, and Recall/Precision COV/AMR are reported as mean and median over molecules (Table 2).

Table 1: Conformation generation under the **4-conformer protocol** (default) across four environments: Vacuum, Docking, Solvation, and Packing. Each model generates 4 conformers per molecule which are compared to the single ground-state reference. We report Recall and Precision COV (↑, mean) and AMR (↓, mean and median) at environment-specific RMSD thresholds: 0.5 Å for Solvation, 0.75 Å for Vacuum and Packing, and 2.0 Å for Docking. For Packing, we additionally report symmetry-aware RMSD (sRMSD, ↓, mean). Per-molecule COV is quantized (Recall $\in \{0, 1\}$; Precision $\in \{0, 0.25, 0.5, 0.75, 1\}$), so we report mean COV only. Best results within each environment are **bolded**. For Vacuum, models are trained and evaluated on the cleaned GEOM-DRUGS dataset (Nikitin et al., 2025), which fixes valency, bond-order, and force-field inconsistencies in the original. EVA-Flow is compared to retrained ETFlow on all environments, and to published baselines for Docking.

| Environment | Setup | Recall | | | Precision | | | sRMSD↓ |
| | | COV↑ | AMR↓ | | COV↑ | AMR↓ | | |
| | | mean | mean | median | mean | mean | median | |
| --- | --- | --- | --- | --- | --- | --- | --- | --- |
| Vacuum | Pretrain+Unified | **0.747** | 0.468 | 0.144 | 0.463 | 0.969 | 0.900 | – |
| | Pretrain+Individual | 0.733 | 0.481 | 0.146 | 0.454 | 0.980 | 0.925 | – |
| | Unified | 0.740 | **0.456** | 0.131 | 0.463 | **0.954** | **0.879** | – |
| | Individual | 0.728 | 0.487 | 0.134 | 0.460 | 0.978 | 0.911 | – |
| | ETFlow (retrained) | 0.725 | 0.488 | **0.130** | **0.467** | 0.972 | 0.919 | – |
| Docking | TankBind | 0.204 | – | – | – | – | – | – |
| | GNINA | 0.321 | – | – | – | – | – | – |
| | DiffDock | 0.382 | – | – | – | – | – | – |
| | DiffDock-L | 0.430 | – | – | – | – | – | – |
| | Pretrain+Unified | **0.684** | 1.899 | **1.257** | 0.512 | 2.471 | 1.864 | – |
| | Pretrain+Individual | 0.656 | 1.980 | 1.463 | 0.458 | 2.581 | 2.076 | – |
| | Unified | 0.674 | **1.889** | 1.303 | **0.536** | **2.391** | **1.732** | – |
| | Individual | 0.553 | 2.458 | 1.833 | 0.378 | 3.135 | 2.387 | – |
| | ETFlow (retrained) | 0.391 | 2.768 | 2.288 | 0.242 | 3.580 | 2.958 | – |
| Solvation | Pretrain+Unified | **0.949** | **0.153** | **0.072** | **0.780** | **0.367** | **0.232** | – |
| | Pretrain+Individual | 0.936 | 0.157 | 0.087 | 0.693 | 0.431 | 0.334 | – |
| | Unified | 0.930 | 0.166 | 0.083 | 0.769 | 0.370 | 0.249 | – |
| | Individual | 0.414 | 0.580 | 0.568 | 0.177 | 0.961 | 0.851 | – |
| | ETFlow (retrained) | 0.605 | 0.475 | 0.366 | 0.264 | 0.824 | 0.786 | – |
| Packing | Pretrain+Unified | **0.644** | **0.664** | **0.509** | 0.410 | 1.073 | 1.008 | 0.055 |
| | Pretrain+Individual | 0.633 | 0.672 | 0.529 | 0.400 | 1.087 | 1.022 | 0.055 |
| | Unified | 0.640 | 0.682 | 0.527 | **0.414** | **1.067** | **0.997** | **0.054** |
| | Individual | 0.512 | 0.819 | 0.734 | 0.303 | 1.222 | 1.161 | 0.064 |
| | ETFlow (retrained) | 0.474 | 0.878 | 0.794 | 0.274 | 1.274 | 1.203 | 0.069 |

**Evaluation metrics**. We evaluate generated conformers against ground-truth conformers using distance-based RMSD. We report the Recall and Precision Coverage (COV, ↑) and average minimum RMSD (AMR, ↓) (Section D). For crystal packing, we also align the generated conformation with the ground-truth one and compute a symmetry-aware RMSD (sRMSD). This value reflects the minimal atomic displacement after accounting for lattice symmetry and periodicity.

## 4.3 Results

Results under the default 4-conformer protocol are summarized in Table 1 and Figure 3; for a fair comparison we retrain ETFlow on the cleaned GEOM-DRUGS dataset (Nikitin et al., 2025) and use it as our primary

Table 2: Conformation generation for Vacuum under the **2K protocol**: each model generates $2K$ conformers per molecule, where $K$ is the number of reference conformers. We report mean and median Recall and Precision COV (↑) and AMR (↓) at RMSD threshold 0.75 Å. Best results are **bolded**. The models are trained on the original GEOM-DRUGS dataset and evaluated on its full test set. Literature numbers for GeoDiff, GeoMol, Torsional Diffusion, MCF-L, and ETFlow (reported) are taken from Table 1 of Hassan et al. (2024); ETFlow (reported) refers to the ET-Flow base (8.3M) variant. ETFlow (retrained) and EVA-Flow are trained on all available conformers per molecule on the original GEOM-DRUGS and evaluated on the same test set.

| Setup | Recall | | | | Precision | | | |
| --- | --- | --- | --- | --- | --- | --- | --- | --- |
| | COV↑ | | AMR↓ | | COV↑ | | AMR↓ | |
| | mean | median | mean | median | mean | median | mean | median |
| GeoDiff | 0.421 | 0.378 | 0.835 | 0.809 | 0.249 | 0.145 | 1.136 | 1.090 |
| GeoMol | 0.446 | 0.414 | 0.875 | 0.834 | 0.430 | 0.364 | 0.928 | 0.841 |
| Torsional Diff. | 0.727 | 0.800 | 0.582 | 0.565 | 0.552 | 0.569 | 0.778 | 0.729 |
| MCF-L | **0.847** | **0.922** | **0.390** | **0.247** | 0.668 | 0.713 | 0.618 | 0.530 |
| ETFlow (reported) | 0.795 | 0.846 | 0.452 | 0.419 | **0.744** | **0.810** | **0.541** | **0.470** |
| ETFlow (retrained) | 0.712 | 0.742 | 0.562 | 0.530 | 0.704 | 0.758 | 0.594 | 0.520 |
| EVA-Flow (Pretrain+Unified) | 0.721 | 0.756 | 0.550 | 0.527 | 0.707 | 0.781 | 0.589 | 0.510 |

Table 3: Visualization of reference and generation conformations in Pretrain+Unified setup for shared molecules in Solvation-Packing and Solvation-Docking. The conformers are aligned to the reference of Solvation.

| Solvation-Packing | | | | Solvation-Docking | | | |
| --- | --- | --- | --- | --- | --- | --- | --- |
| ref_solv | gen_solv | ref_pack | gen_pack | ref_solv | gen_solv | ref_dock | gen_dock |

baseline. To compare against published Vacuum baselines that were trained on the *original* GEOM-DRUGS dataset under the 2K protocol, we additionally train EVA-Flow and ETFlow on the same data and report results in Table 2: EVA-Flow improves over the retrained ETFlow on every metric and narrows the Recall gap with the 20×-larger MCF-L (242M vs. 12.1M parameters) while surpassing it on Precision.

**Pretraining improves generalization.** Across all environments, pretrained setups consistently outperform their non-pretrained counterparts (Table 1, Figure 3). The effect is most pronounced for data-scarce environments (Solvation, Packing, Docking) and less so for Vacuum, whose large training set already dominates pretraining.

**Unified training outperforms individual models.** Pretrain+Unified and Unified setups consistently surpass Individual finetuning, and Pretrain+Unified also outperforms Pretrain+Individual (Table 1). Cross-environment training enables the model to share chemical knowledge, improving accuracy even on environments with limited data. A data-volume control (total molecule exposure and a balanced-sampling experiment, Appendix I) attributes this gain to cross-environment transfer rather than data volume.

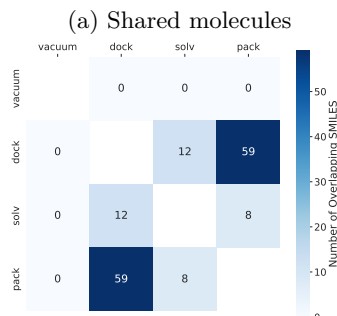
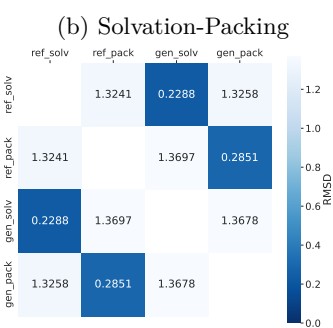
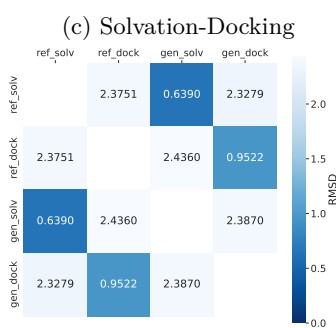

Figure 4: Shared-molecule analysis across environments. (a) Pairwise counts of molecules shared between environments. Darker cells indicate more molecules. (b, c) Pairwise heavy-atom RMSD (Å) between reference and generated conformers for molecules shared across Solvation-Packing and Solvation-Docking. We report the mean RMSD after alignment for: ref_A ↔ ref_B, gen_A ↔ ref_A, gen_A ↔ ref_B, gen_B ↔ ref_A, gen_B ↔ ref_B, and gen_A ↔ gen_B (A/B = the two environments). Darker cells indicate lower RMSD (better agreement).

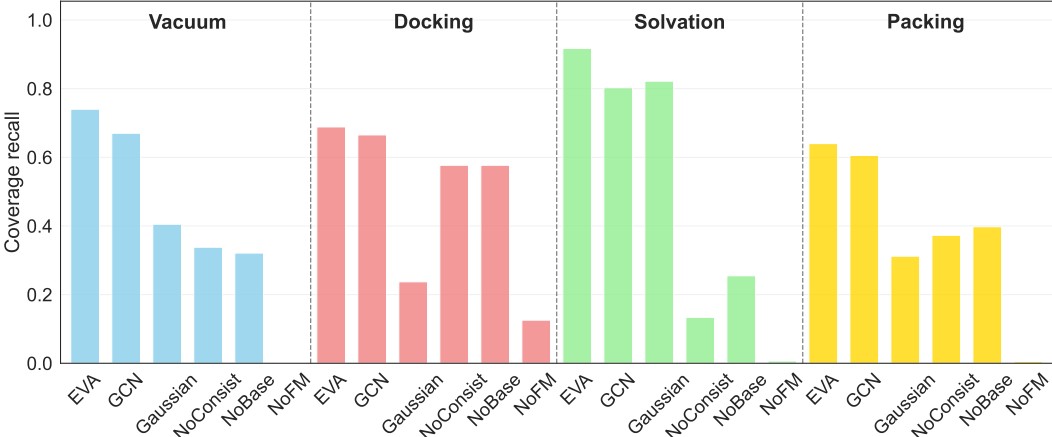

Figure 5: Recall COV for all environments (Vacuum, Docking, Solvation, Packing) using the Unified setup for small model. 'EVA' refers to standard setup; 'GCN' uses non-equivariant networks for the encoder and base; 'Gaussian' parameterizes the base distribution as a Gaussian distribution instead of Harmonic; 'NoConsist' is removing latent consistency loss; 'NoBase' is for replacing the learned base distribution network for initial positions $x_0$ with a Harmonic distribution determined only by the graph; 'NoFM' is removing the FM decoder and using a regression decoder.

## 4.4 Analysis

To test whether EVA-Flow truly conditions on the environment rather than memorizing a single geometry, we examine molecules that appear in multiple environments (Figure 4a). The overlaps are small but sufficient for meaningful comparison (e.g., 12 molecules shared between Solvation and Docking). Example visualizations are in Tables 3 and 11.

**Environment awareness when references differ.** For Solvation–Docking (Figure 4c), reference conformers of the same molecule differ substantially (2.33 Å). Each generated conformer is close to its own environment's reference (Solvation: 0.64 Å; Docking: 0.95 Å), while cross-environment RMSD remains large (> 2.3 Å). Similar patterns hold for Solvation–Packing (Figure 4b), confirming that the model adapts conformations to the specified environment. Zeroing out domain-specific environment features at inference degrades performance in every environment (Appendix G), confirming the model actively uses each environment representation.

**When environments agree, generations agree**. For Packing–Docking (Figure 9b), the two references already agree (0.52 Å), and generated conformers across environments are correspondingly close (0.81 Å). These complementary patterns that divergent generations when references differ, convergent when they agree confirm genuine environment awareness.

### 4.5 Ablation Studies

We ablate key components using the Unified setup with the small model (Figure 5). The full model (EVA) is defined in Sections 3.2 and 3.3. Variants: **GCN** replaces SE(3)-equivariant modules with graph convolutional networks; **Gaussian** uses a diagonal Gaussian base instead of the Harmonic distribution; **NoConsist** removes the latent consistency loss; **NoBase** replaces the learned base network with the graph-topology Harmonic prior $\mathcal{N}(\mathbf{0}, L^\dagger)$; **NoFM** replaces the FM decoder with a regression decoder.

Figure 5 reports Recall COV of these ablations for all environments. The full model achieves the highest recall in all settings; the complete Precision and AMR metrics are reported in Appendix H. The ablation of the FM decoder consistently underperforms across all environments, because regression tends to learn the conditional mean of a multi-modal conformer distribution, leading to blurred geometry and lower coverage. Removing the base distribution network (NoBase) causes a large performance drop, indicating that a learned, environment-aware base distribution is essential for modeling molecule–environment interactions. This drop reflects the learned posterior's role as a training scaffold rather than an inference-time distribution gap: at convergence the posterior matches the harmonic prior used at inference (Appendix J). Eliminating the latent consistency loss (NoConsist) produces a consistent reduction in Recall COV, suggesting that latent consistency stabilizes training and provides effective regularization. Replacing the harmonic base with a Gaussian one (Gaussian) also degrades performance, highlighting the importance of incorporating graph geometry through the harmonic formulation. Finally, using non-equivariant encoders and base networks (GCN) leads to inferior results, demonstrating the benefit of SE(3)-equivariant architectures for conformer generation.

## 5 Conclusion

We introduced EVA-Flow, a unified framework for environment-aware molecular conformation generation. By coupling a VAE encoder with an environment-conditioned flow matching decoder and a learned Harmonic base distribution, EVA-Flow generates accurate conformers across vacuum, protein–ligand docking, solvation, and crystal packing within a single model. Analysis of shared molecules confirms that EVA-Flow produces distinct, environment-specific conformations rather than collapsing to a single geometry.

Several directions remain open. More expressive backbone architectures and scaling to larger models could improve accuracy, particularly for complex environments. Our evaluation covers four environments; extending to settings such as surface adsorption or solid–liquid interfaces will require new benchmarks and poses an exciting avenue for future work.

## Broader Impact Statement

EVA-Flow is intended to support molecular modeling in chemistry, materials science, and drug discovery. More accurate conformation generation could reduce the computational cost of downstream simulations and facilitate scientific discovery. Like other molecular generative and predictive tools, however, the method may produce inaccurate geometries and should not replace experimental validation or high-accuracy physical simulation in safety-critical applications. We recommend that generated structures be validated with appropriate physical calculations and domain expertise before downstream use.

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

## A  Data Preparation

**GEOM QM9 and DRUGS.**  We follow the data processing pipeline of ETFlow (Hassan et al., 2024) and adopt the same train/validation/test splits. For GEOM-DRUGS, We applied the data cleaning strategies reported in (Nikitin et al., 2025) that fixes valency, bond-order, and force-field inconsistencies in the original datasets. For each molecule, we retain the top 30 conformers ranked by Boltzmann weight. Each molecule

is stored as a PyTorch Geometric object containing atomic numbers, 3D positions, and a canonical SMILES string. The QM9 training set contains 106,587 molecules (794,960 conformations) and the validation set 13,323 molecules (99,945 conformations); the test set has 1,000 molecules. The DRUGS training set contains 272,641 molecules (5,770,377 conformations) and the validation set 30,405 molecules (715,592 conformations); the test set has 1,000 molecules.

**SPICE and CREST-relaxed SPICE.** We use the PubChem subset of SPICE (Eastman et al., 2023), which contains small drug-like molecules (18–50 atoms; elements Br, C, Cl, F, H, I, N, O, P, S).

Because the SPICE conformers are not fully relaxed, we construct a complementary relaxed set via CREST (Pracht et al., 2024). For each molecule we (i) generate 50 initial conformers with RDKit `EmbedMultipleConfs` (pruning threshold 0.01 Å, MMFF optimization), (ii) deduplicate with `GetBestRMS` (threshold 0.1 Å), (iii) refine the ten lowest-energy conformers with extended tight binding (xTB) (Grimme et al., 2017; Friede et al., 2024), and (iv) run CREST v3.0.2 from the lowest-xTB-energy conformer with a 6 kcal/mol cutoff. Molecules with cis/trans stereo-SMILES were excluded because the stereochemistry is not preserved through the pipeline.

For both SPICE and CREST-relaxed SPICE, we align atom ordering between the dataset coordinates and the RDKit molecule built from the canonical SMILES using `GetSubstructMatch`. We split 0.8/0.1/0.1 into train/validation/test. The training set contains 19,198 molecules (959,802 SPICE conformers; 2,203,392 CREST conformers), the validation set 2,508 molecules (125,370; 278,673), and the test set 2,941 molecules.

**OE62.** OE62 (Stuke et al., 2020) provides PBE-relaxed geometries of crystal-forming organic molecules. We read the SMILES and coordinates, align atom ordering via `GetSubstructMatch` as above, and split 0.8/0.1/0.1. The training set contains 28,038 molecules, the validation set 4,297, and the test set 3,933 (one conformation each).

**PDBBind.** We use PDBBind-v2020 (Liu et al., 2015). For each protein–ligand pair, we extract the ligand from the SDF file, add hydrogens, and generate a canonical SMILES. Atom ordering between the SDF-derived and SMILES-derived RDKit molecules is aligned via `GetSubstructMatch`.

The protein binding pocket serves as the environment graph. We parse the pocket PDB file and retain only heavy atoms, adding edges between atoms within the same residue. Node features follow ETFlow conventions.

We use the original PDBBind splits. After filtering invalid ligands, the training set has 9,940 examples, the validation set 614, and the test set 218 (one ligand conformation each).

**Aquamarine (AQM).** Aquamarine (Medrano Sandonas et al., 2024) contains conformations relaxed in implicit water. We build RDKit molecules from SMILES, add hydrogens, and align atom ordering as above. The environment graph uses a water molecule with four normalized solvation descriptors appended to node features: cavity surface area (sCAV), cavity volume (vCAV), free energy in electrolyte (eSOLV), and non-electrostatic free energy (eNELEC). Water atom coordinates are set to zero.

After filtering, we split 0.8/0.1/0.1. The training set has 1,276 molecules, the validation set 175, and the test set 157 (one conformation each).

**Crystallography Open Database (COD).** The COD (Gražulis et al., 2009; 2012) is an open-access collection of crystal structures. We select organic molecular crystals by applying the following filters: the structure must (i) contain organic molecules, (ii) not be marked as disordered (Groom et al., 2016), (iii) not be missing hydrogen, (iv) have a valid SMILES (Weininger, 1988) with a single fragment, (v) pass structural validation (Ong et al., 2013), and (vi) contain exactly one formula unit in the asymmetric unit ($Z' = 1$).

We recover the molecular topology with RDKit maximum-common-substructure matching (`FindMCS`) (Landrum & contributors, 2025) rather than the often-unreliable COD SMILES, and extract the space group, Wyckoff positions, and Cartesian coordinates for each formula unit using `pymatgen` (Ong et al., 2013). We

process structures into PyTorch Geometric objects, aligning atom ordering via RDKit `GetSubstructMatch`. Lattice parameters $(a, b, c, \alpha, \beta, \gamma)$ and the space group number are included as features.

After filtering, we split 0.8/0.1/0.1. The training set has 20,765 molecules, the validation set 3,239, and the test set 3,235 (one conformation each).

## B    Distribution of Number of Atoms and Number of Rotatable Bonds

To characterize the intrinsic difficulty of conformer generation across datasets, we report histograms of molecular size (number of atoms, $N_{\text{atoms}}$) and flexibility (number of rotatable bonds, $N_{\text{rot}}$) for each dataset (Figures 6 and 7). We compute $N_{\text{rot}}$ using `rdMolDescriptors.CalcNumRotatableBonds` from RDKit with its default definition (non-ring single bonds between non-terminal heavy atoms).

Across the pretraining sets (GEOM Drugs, GEOM QM9, SPICE, CREST-relaxed SPICE, and OE62), $N_{\text{atoms}}$ concentrates in the mid-size regime with means in the $\sim 17$–$44$ range, while $N_{\text{rot}}$ is typically modest (most molecules have fewer than 15 rotatable bonds).

For the finetuning sets (GEOM Drugs, PDBBind, AQM, and COD), the atom-count distribution shifts larger with means in the $\sim 38 - 64$ range. In particular, the PDBBind dataset exhibits a long tail (very large molecules), indicating increased geometric and torsional complexity.

## C    Training Details and Hyperparameters

All models are trained on 8 NVIDIA A100 GPUs with AdamW and a cosine-annealing learning rate schedule with linear warmup. Table 4 lists the per-size hyperparameters.

**Pretraining.** We train for 800 epochs, resampling 100K examples per epoch from the pretraining pool with mixing probabilities 0.6 (GEOM-DRUGS), 0.2 (OE62), 0.1 (GEOM-QM9), 0.05 (SPICE), and 0.05 (CREST).

**Unified finetuning.** Starting from the pretrained checkpoint, we finetune for 800 epochs, resampling 100K examples per epoch with probabilities 0.6 (Vacuum / GEOM-DRUGS), 0.2 (Packing / COD), 0.15 (Docking / PDBBind), and 0.05 (Solvation / AQM).

**Individual finetuning.** We finetune on each environment's full training set for 800 epochs using the same optimizer and schedule.

### C.1    Training, Inference, and Reproducibility Details

**Optimization and schedule.** We use AdamW with $(\beta_1, \beta_2) = (0.95, 0.999)$, weight decay $10^{-8}$, and gradient-norm clipping at 1.0. The learning rate follows a cosine-annealing-with-warmup schedule: a linear warmup of 31,250 steps to a peak of $5 \times 10^{-4}$, then cosine decay over a 312,500-step cycle to a minimum of $10^{-8}$.

**Data splits, validation, and checkpointing.** Each environment uses fixed train/validation/test splits. We validate every epoch and monitor the total validation loss (`val/total_loss`); the model checkpoint with the lowest validation loss is used for all reported results.

**Random seeds.** Training is seeded with seed 42. Inference uses the SDE sampler, which is stochastic and not explicitly seeded; we therefore report variance across independent sampling runs where relevant (e.g., the all-conformer ETFlow baseline is reported as mean $\pm$ std over six runs).

**Inference sampler and noise.** Unless otherwise noted, all reported numbers use the SDE sampler with 50 integration steps, additive-noise scale std = 1.0, churn $s_{\text{churn}} = 1.0$, and a stochastic window $t \in [10^{-4}, 0.9999]$; the initial state $x_0$ is drawn from the harmonic prior with no additional perturbation. The deterministic ODE sampler is also supported and trades a small amount of Recall COV for speed (Vacuum: 0.737 ODE vs. 0.747 SDE).

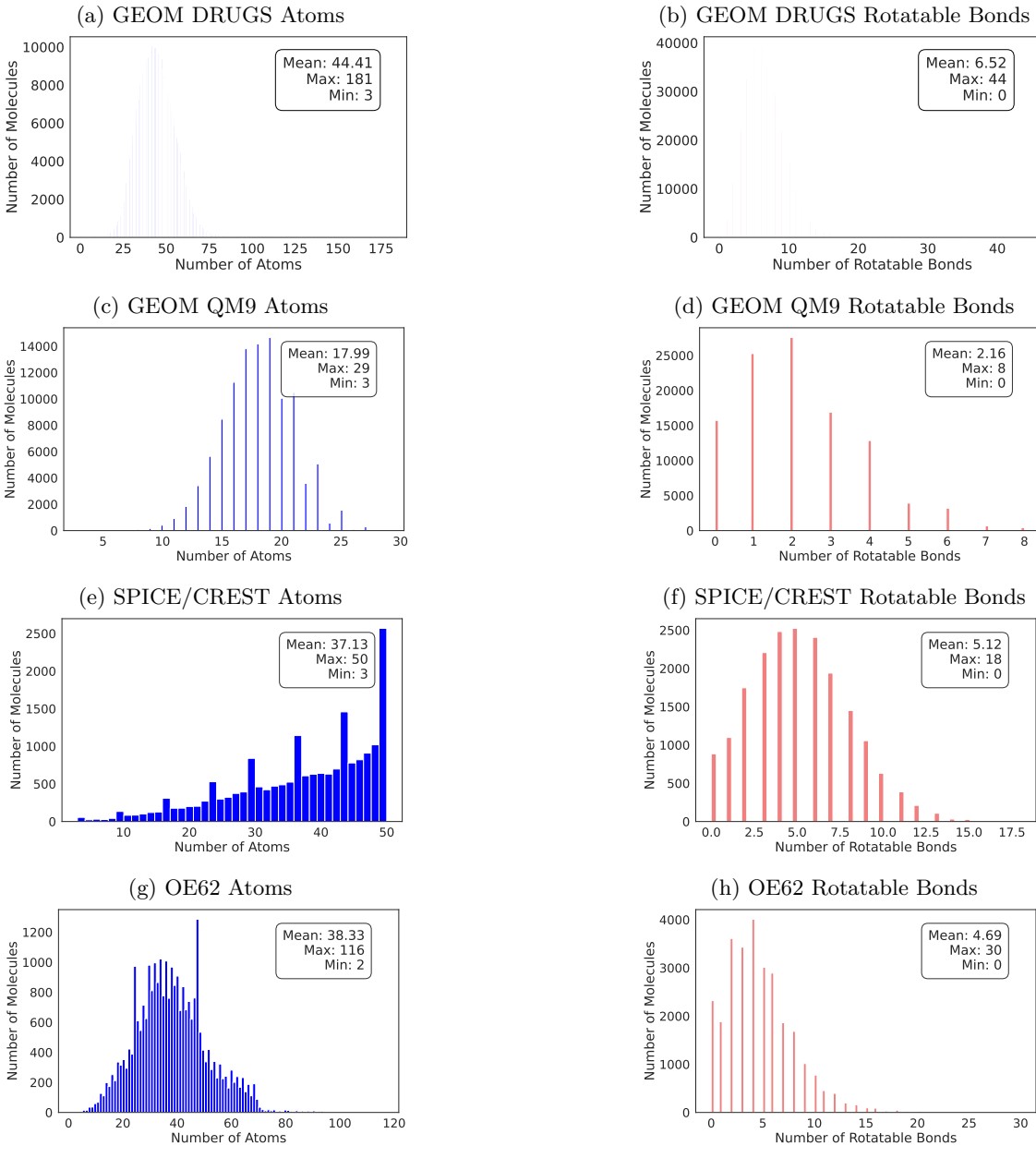

Figure 6: The histograms of the number of atoms and the number of rotatable bonds in the pretraining datasets, including GEOM DRUGS, GEOM QM9, SPICE, CREST-relaxed SPICE, and OE62.

**Inference time.** Table 5 reports per-molecule inference time on a single A100 GPU (GEOM-DRUGS test set). EVA-Flow is faster than the retrained ETFlow at the same step count, as the learned base distribution provides a better initialization.

**Compute.** Models are trained on $8\times$ NVIDIA A100 GPUs. Each 800-epoch stage (pretraining and unified finetuning) takes roughly 5 minutes per epoch, i.e. approximately 67 wall-clock hours ($\sim$530 A100-GPU-hours); individual finetuning is run per environment on its full training set.

**Dataset preprocessing.** Each dataset is processed by a dedicated script that will be released in the future. Molecular graphs and features are built with `RDKit` and `datamol`; crystal lattices and space groups

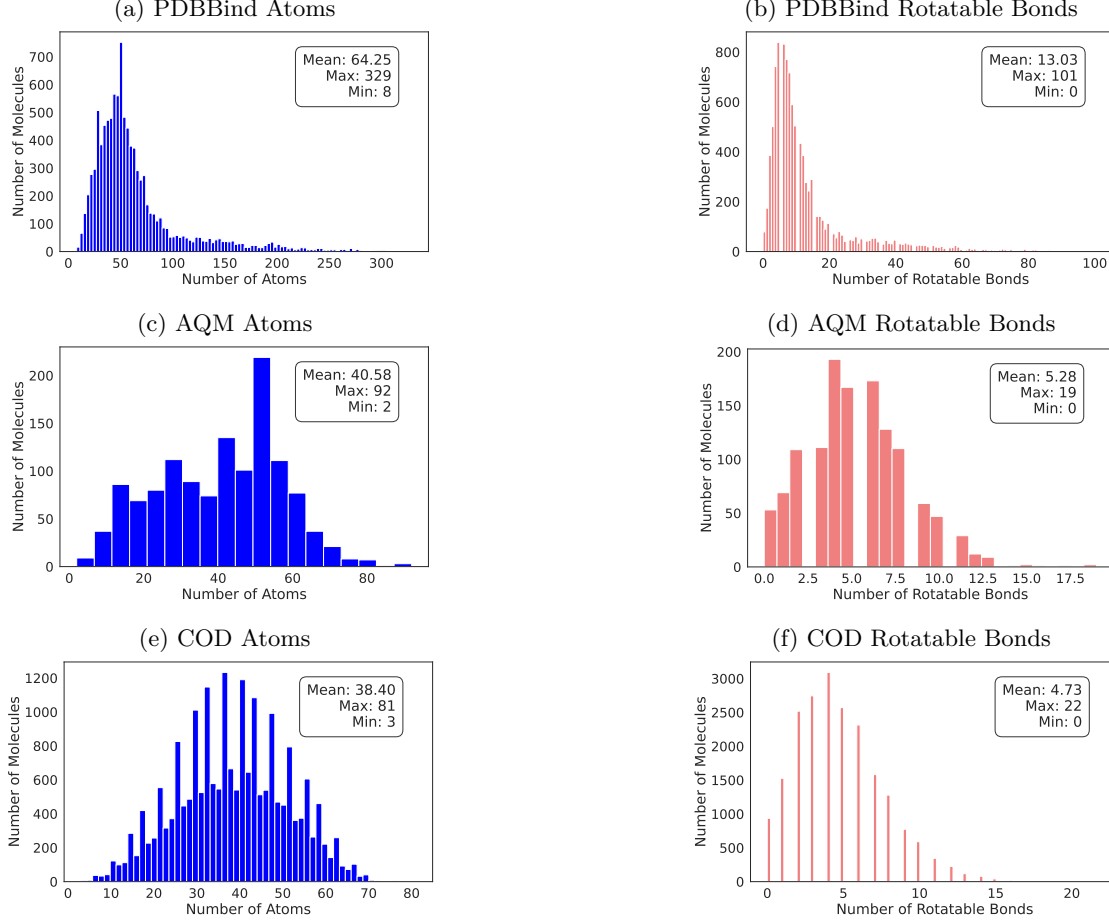

Figure 7: The histograms of the number of atoms and the number of rotatable bonds in the finetuning datasets, including PDBBind, AQM, and COD. The GEOM DRUGS dataset is the same as the one in the pretraining datasets (Figure 6).

for Packing use `pymatgen`, with `OpenBabel` for bond perception from CIF files; numerical processing uses `NumPy` and `SciPy`.

## D   Evaluation Metric Definition

We compute average minimum RMSD (AMR) and Coverage (COV) Recall and Precision to assess the performance of molecular conformer generation following the approaches of (Ganea et al., 2021; Xu et al., 2022; Jing et al., 2022). Recall measures the extent to which the generated conformers capture the ground-truth conformers, while Precision indicates the proportion of generated conformers that match the ground-truth conformers. Using $C_g$ to denote the set of generated conformations, and $C_r$ the set of reference conformations, we calculate these metrics using the following equations:

$$\text{AMR-R}(C_g, C_r) = \frac{1}{|C_r|} \sum_{\mathbf{R} \in C_r} \min_{\hat{\mathbf{R}} \in C_g} \text{RMSD}(\mathbf{R}, \hat{\mathbf{R}}) \tag{19}$$

$$\text{COV-R}(C_g, C_r) = \frac{1}{|C_r|} |\{\mathbf{R} \in C_r | \text{RMSD}(\mathbf{R}, \hat{\mathbf{R}}) < \delta, \hat{\mathbf{R}} \in C_g\}| \tag{20}$$

$$\text{AMR-P}(C_r, C_g) = \frac{1}{|C_g|} \sum_{\hat{\mathbf{R}} \in C_g} \min_{\mathbf{R} \in C_r} \text{RMSD}(\hat{\mathbf{R}}, \mathbf{R}) \tag{21}$$

Table 4: Hyperparameters for EVA-Flow.

| Hyperparameter | Value |
|---|---|
| Batch size | 32 |
| FM decoder layers | 20 |
| Hidden channels | 160 |
| Attention heads | 8 |
| Latent dim ($d_z$) | 128 |
| Environment dim ($d_e$) | 128 |
| Encoder layers | 3 |
| Base layers | 3 |
| Environment layers | 3 |
| Cutoff range | [0.0, 10.0] Å |
| Activation | SiLU |
| RBF type / count | ExpNorm / 64 |
| Optimizer | AdamW |
| AdamW ($\beta_1, \beta_2$) | (0.95, 0.999) |
| Max LR | $5 \times 10^{-4}$ |
| Min LR | $10^{-8}$ |
| Weight decay | $10^{-8}$ |
| Warmup steps | 31,250 |
| Cosine cycle length | 312,500 steps |
| Gradient clip (norm) | 1.0 |
| Random seed | 42 |
| Inference sampler / steps | SDE / 50 |
| SDE noise ($s_{\text{churn}}$, std) | (1.0, 1.0) |
| GPUs | 8× A100 |
| Total parameters | 12.1M |

Table 5: Per-molecule inference time on a single A100 GPU (GEOM-DRUGS, 1,000 molecules). EVA-Flow uses the Pretrain+Unified setup.

| Method | Sampler | Steps | R-COV ↑ | Time (s/mol) ↓ |
|---|---|---|---|---|
| ETFlow (retrained) | SDE | 50 | 0.725 | 0.309 |
| EVA-Flow (P+U) | SDE | 50 | 0.737 | 0.266 |
| EVA-Flow (P+U) | SDE | 100 | 0.752 | 0.530 |

$$\text{COV-P}(C_r, C_g) = \frac{1}{|C_g|} |\{\hat{\mathbf{R}} \in C_g | \text{RMSD}(\hat{\mathbf{R}}, \mathbf{R}) < \delta, \mathbf{R} \in C_r\}| \tag{22}$$

A lower AMR score signifies improved accuracy, while a higher COV score reflects greater diversity in the generative model. The threshold $\delta$ is set to 0.5 Å for Solvation, 0.75 Å for Vacuum (Hassan et al., 2024) and Packing, and 2.0 Å for Docking (Corso et al., 2022).

# E   Additional Results

## E.1   Additional Precision Results

Precision Coverage plots are shown in Figures 8.

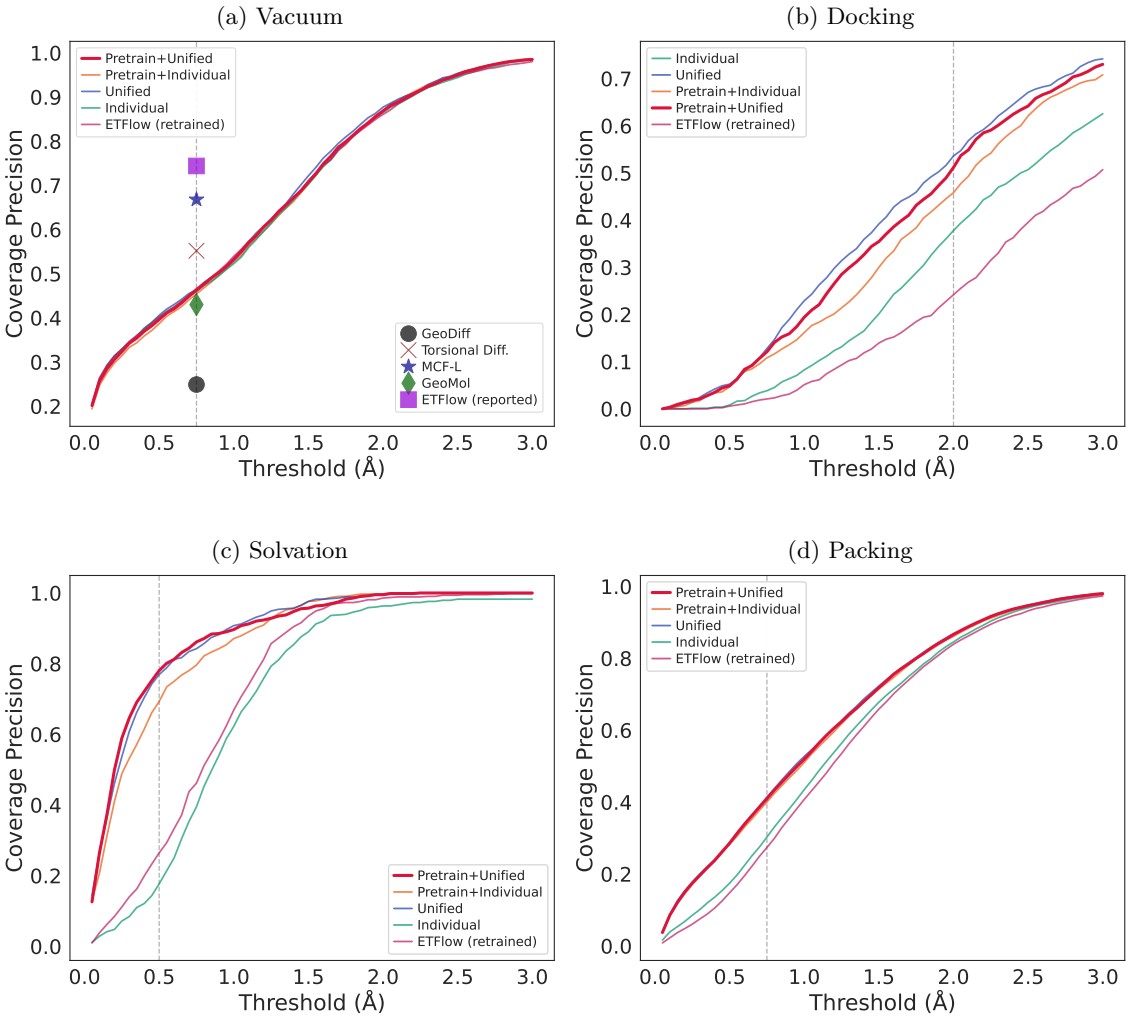

Figure 8: Precision Coverage for all four environments: Vacuum, Docking, Solvation, and Packing.

## F    Physical Validity of Generated Conformers

Coverage and AMR measure geometric similarity to reference conformers but not energetic plausibility. To assess physical validity directly, we relax all 4,000 generated Vacuum conformers (1,000 molecules $\times$ 4 conformers) with UMA (Wood et al., 2026), a large-scale pretrained molecular potential, and measure the energy change and geometric shift upon relaxation (Table 6). The very small energy changes and atomic displacements indicate that EVA-Flow generates conformers that are already near local energy minima according to a state-of-the-art potential.

## G    Environment-Graph Sensitivity

To verify that the model actively uses each environment representation, we zero out domain-specific environment features at inference time and measure the performance change (Table 7). **NoSolvDesc** zeros the four solvation descriptors; **NoLattice** zeros the seven lattice/space-group parameters; **NoPocket** zeros the entire environment embedding for docking. All three ablations degrade performance, confirming that the model relies on each environment component, and the docking pocket is the most critical.

Table 6: UMA relaxation of EVA-Flow Vacuum conformers (4,000 conformers from 1,000 molecules).

| Metric | Value |
|---|---|
| UMA relaxation converged | 99.9% (3,995/4,000) |
| Mean $\Delta E$ after relaxation | 0.302 kcal/mol |
| Median $\Delta E$ after relaxation | 0.115 kcal/mol |
| Fraction with $\Delta E < 1$ kcal/mol | 97.1% |
| Mean RMSD shift during relaxation | 0.056 Å |
| Fraction with RMSD shift $< 0.1$ Å | 84.2% |

Table 7: Environment-feature sensitivity (inference-time zeroing, no retraining). Thresholds: Solvation 0.5 Å, Packing 0.75 Å, Docking 2.0 Å.

| Environment | Ablation | R-COV ↑ | R-AMR ↓ | P-COV ↑ | P-AMR ↓ |
|---|---|---|---|---|---|
| Solvation | Full | **0.949** | **0.153** | **0.780** | **0.367** |
| | NoSolvDesc | 0.905 | 0.191 | 0.734 | 0.399 |
| Packing | Full | **0.644** | **0.664** | **0.410** | **1.073** |
| | NoLattice | 0.575 | 0.762 | 0.360 | 1.154 |
| Docking | Full | **0.684** | **1.899** | **0.512** | **2.471** |
| | NoPocket | 0.400 | 2.590 | 0.257 | 3.197 |

## H   Full Ablation Metrics

Figure 5 reports Recall COV for the ablations. Table 8 provides the complete metrics (Recall and Precision COV and AMR) across all four environments. The full model is best or near-best on every metric and environment, and the trends match the Recall-COV results: removing the FM decoder (NoFM) or the learned base (NoBase) is most damaging, followed by removing the latent-consistency loss (NoConsist), the Harmonic base (Gaussian), and SE(3)-equivariance (GCN).

## I   Data Exposure and Balanced Sampling

To test whether the Unified advantage stems from cross-environment transfer rather than data volume, we analyze total molecule exposure and run a balanced-sampling control. Table 9 (top) counts the molecule samples each setup sees: Unified uses epoch-level resampling (epoch size 100K, probabilities 60/20/5/15%, capped at dataset size), while Individual models pass over their full dataset each epoch. Unified sees *fewer* DRUGS molecules than Individual (42.0M vs. 54.5M) yet outperforms it; for the small-data environments the Individual models already overfit (validation loss plateaus by epoch $\sim$30–50), so additional repetitions alone cannot explain the gains.

Table 9 (bottom) compares balanced (25/25/25/25%) against the default weighted sampling, both from the same pretrained checkpoint. Solvation and Docking see the *same* number of molecules per epoch under both schemes (capped at dataset size), yet Solvation drops 10.7% when the DRUGS allocation falls from 60% to 25%, indicating that DRUGS data benefits Solvation through cross-environment transfer in the shared backbone, not through Solvation data volume.

## J   Train–Test Consistency of the Base Distribution

At inference the initial coordinates are drawn from the simple harmonic prior rather than the learned posterior. We verify that this introduces negligible mismatch. First, the KL divergence $\mathrm{KL}(q_{\mathrm{posterior}} \parallel p_{\mathrm{prior}})$ at convergence is 0.036 nats (validation; 0.026 train). Second, comparing the learned posterior's eigenvalues to the harmonic prior's on 200 test molecules gives an eigenvalue ratio with median 1.0000 and 100% of

Table 8: Full ablation metrics (Unified setup, small model). Thresholds: Vacuum/Packing 0.75 Å, Solvation 0.5 Å, Docking 2.0 Å. Best per metric/environment in **bold**.

| Metric | Setup | Vacuum | Docking | Solvation | Packing |
|---|---|---|---|---|---|
| R-COV ↑ | EVA (Full) | **0.740** | **0.674** | **0.930** | **0.640** |
| | GCN | 0.670 | 0.665 | 0.803 | 0.605 |
| | Gaussian | 0.405 | 0.237 | 0.822 | 0.312 |
| | NoConsist | 0.338 | 0.577 | 0.134 | 0.373 |
| | NoBase | 0.321 | 0.577 | 0.255 | 0.398 |
| | NoFM | 0.000 | 0.126 | 0.006 | 0.005 |
| P-COV ↑ | EVA (Full) | **0.463** | **0.536** | **0.769** | **0.414** |
| | GCN | 0.396 | 0.490 | 0.497 | 0.384 |
| | Gaussian | 0.241 | 0.179 | 0.618 | 0.201 |
| | NoConsist | 0.184 | 0.379 | 0.102 | 0.206 |
| | NoBase | 0.177 | 0.376 | 0.140 | 0.220 |
| | NoFM | 0.000 | 0.057 | 0.003 | 0.002 |
| R-AMR ↓ | EVA (Full) | **0.456** | **1.889** | **0.166** | **0.682** |
| | GCN | 0.623 | 2.153 | 0.285 | 0.714 |
| | Gaussian | 1.078 | 4.528 | 0.291 | 1.248 |
| | NoConsist | 1.108 | 2.221 | 0.896 | 0.990 |
| | NoBase | 1.113 | 2.160 | 0.790 | 0.963 |
| | NoFM | 2.136 | 3.536 | 1.735 | 2.214 |
| P-AMR ↓ | EVA (Full) | **0.954** | **2.391** | **0.370** | **1.067** |
| | GCN | 1.117 | 2.937 | 0.593 | 1.117 |
| | Gaussian | 1.551 | 5.414 | 0.492 | 1.601 |
| | NoConsist | 1.569 | 2.941 | 0.921 | 1.400 |
| | NoBase | 1.560 | 2.711 | 0.902 | 1.363 |
| | NoFM | 2.502 | 4.020 | 2.145 | 2.581 |

Table 9: Top: total molecule exposure across training. Bottom: balanced vs. default sampling (Recall COV at the environment threshold).

| Environment | Dataset | Unified/ep | Unified tot. | Indiv./ep | Indiv. tot. |
|---|---|---|---|---|---|
| Vacuum (DRUGS) | 272,641 | 60,000 | 42.0M | 272,641 | 54.5M |
| Packing (COD) | 20,627 | 20,000 | 14.0M | 20,627 | 4.1M |
| Solvation (AQM) | 1,276 | 1,276 | 0.89M | 1,276 | 0.26M |
| Docking (PDB) | 9,940 | 9,940 | 6.96M | 9,940 | 0.99M |

| Environment | Default (60/20/5/15) | Balanced (25/25/25/25) | Change |
|---|---|---|---|
| Vacuum (0.75 Å) | 0.740 | 0.692 | −6.5% |
| Docking (2.0 Å) | 0.684 | 0.670 | −2.0% |
| Solvation (0.5 Å) | 0.949 | 0.847 | −10.7% |
| Packing (0.75 Å) | 0.644 | 0.643 | −0.2% |

modes within 1% of unity (Pearson correlation 1.000000), with posterior mean $\approx 0$: the learned posterior has effectively converged to the prior. Third, a noise-injection sweep that perturbs the harmonic prior $x_0 = x_0^{\mathrm{harm}} + \epsilon \cdot \mathcal{N}(0, I)$ degrades gracefully (Table 10), confirming the flow network corrects moderate base-distribution errors. The large NoBase drop (Table 8) therefore reflects the learned posterior's value as a *training scaffold* (better gradients), not an inference-time distribution gap.

Table 10: Inference noise-injection sweep on Vacuum (Individual model, SDE 50 steps): perturbing the harmonic prior by $\epsilon \cdot \mathcal{N}(0, I)$.

| $\epsilon$ | R-COV $\uparrow$ | R-AMR $\downarrow$ | P-COV $\uparrow$ | P-AMR $\downarrow$ |
|---|---|---|---|---|
| 0.00 (default) | **0.747** | **0.468** | **0.464** | **0.979** |
| 0.01 | 0.738 | 0.470 | 0.455 | 0.983 |
| 0.05 | 0.735 | 0.493 | 0.459 | 0.981 |
| 0.10 | 0.745 | 0.471 | 0.464 | 0.973 |
| 0.20 | 0.730 | 0.486 | 0.452 | 0.985 |
| 0.50 | 0.734 | 0.480 | 0.453 | 0.982 |
| 1.00 | 0.704 | 0.529 | 0.413 | 1.079 |

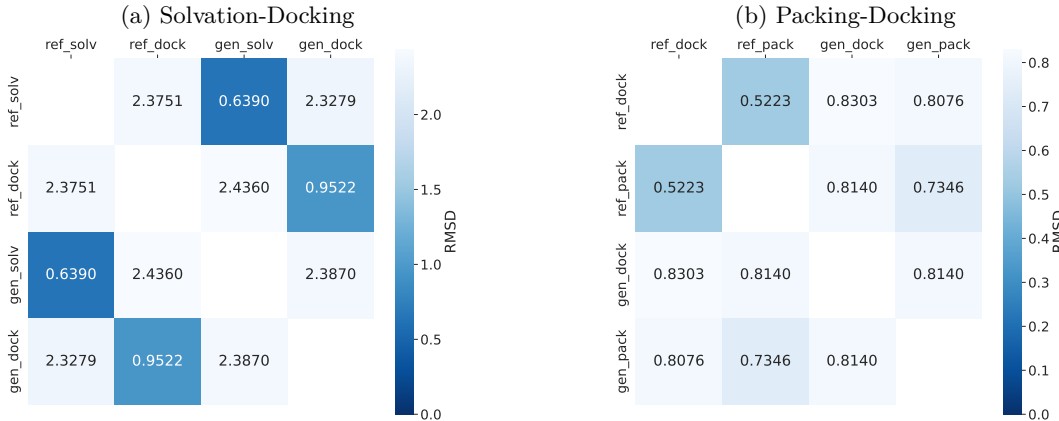

Figure 9: Pairwise heavy-atom RMSD (Å) between reference and generated conformers shared across: (a) Solvation-Docking, and (b) Packing-Docking. In each heatmap we report the mean RMSD after alignment for: ref_A $\leftrightarrow$ ref_B, gen_A $\leftrightarrow$ ref_A, gen_A $\leftrightarrow$ ref_B, gen_B $\leftrightarrow$ ref_A, gen_B $\leftrightarrow$ ref_B, and gen_A $\leftrightarrow$ gen_B (A/B = the two tasks in the panel). Darker cells indicate lower RMSD (better agreement).

## K  Molecule Visualization

The visualization of reference and generated conformers for molecules shared in the Packing-Docking environments.

Table 11: Visualization of reference and generation conformations for the shared molecules in Packing-Docking. The molecules are aligned to the reference conformers of Docking environment.

| Packing-Docking | | | |
|---|---|---|---|
| ref_dock | gen_dock | ref_pack | gen_pack |

