# OpenReview forum: "EVA-Flow: Learning Shared Chemistry for Unified Environment-Aware Molecular Conformation Generation"
_TMLR — Under review for TMLR_

### Review · Reviewer_ZqCB · 2026-07-07

**Summary Of Contributions:**

EVA-Flow is a new framework to condition the generation of a 3D conformation based on the environment. The authors propose a "unified" architecture that uses an environment encoder to condition the flow-based decoder for generating 3D molecular conformations.
- The core idea of conditioning the generation of 3D conformations based on the underling environment is sound and of major interest for several applications.
- The paper is well written but I have some concerns on a few technical aspects that I would like the authors to clarify.
- The unified approach shows empirical evidence of cross-environment transfer
- The authors provide clear details on how to reproduce the results in practice, including dataset, training parameters and processing details

**Audience:**

Yes

**Audience Explanation:**

I think that the core idea of conformation generation conditioned on the environment is of interest for several applications, including drug discovery and protein design. However, I have concerns on the experimental design and the reported results, which I think should be clarified by the authors.

**Broader Impact Concerns:**

The paper addresses ethical concerns related to the developed framework.

**Claims And Evidence:**

No

**Claims Explanation:**

The core idea of conditioning the generation of 3D conformations based on the underling environment is sound and of major interest for several applications. However, I have some concerns that I would like the authors to clarify. Concerns are reported in the "Requested Changes" section below.

**Requested Changes:**

To me it doesn't sound convincing that you retrained ETFlow. Why are the scores presented in the original paper (https://arxiv.org/pdf/2410.22388) not matching the ones you are reporting? They seem to be significantly worse. Can you provide a justification for that?

I see you are writing that you are re-training on a "cleaned GEOM-DRUGS", however, I don't see any reason why this would lead to such a significant drop in performance (see Table 1, AMR recall median or COV recall if compared to ETFlow Table 1 https://arxiv.org/pdf/2410.22388).

Why aren't you reporting the median scores for COV (Table 1), which are usually more informative than the mean ones, especially in this setting where the distribution of the scores can contain some outliers?

In Section 3.2 (point 1), the environment network is not equivariant.  Can you provide a justification for that?

For docking, when including the protein coordinates, the environment embedding changes under rigid rotation even if the pocket is kept fixed. Can you provide an ablation comparing equivariant vs. non-equivariant models?

At inference time, z is sampled independently from N(0, I). Could it be possible that the encoder suffers from posterior collapse and the decoder is just using the information from E and G?

I suspect that the comparison with DiffDock, TankBind and GNINA is not entirely fair, since they predict the translation, rotation, and  conformation of the ligand given a protein structure. On the other hand, EVA-Flow given a pocket predicts just the conformation assuming that the ligand is already positioned correctly. Let me know if I am missing something here. If this is the case, I would suggest removing these comparisons from the paper.

Regarding the Aquamarine dataset, I see that you are using for training 1276 molecules with one conformer each. What's the reasoning behind the choice of using only one conformer per molecule? What's the selection criterion for the conformer that you are using for training?

---

### Review · Reviewer_o9Kd · 2026-07-07

**Summary Of Contributions:**

The central contribution of the paper is to formulate conformer generation as learning p(x∣G,E), where the generated 3D coordinates depend not only on the molecular graph G but also on an explicit representation of the environment E. Rather than training separate conformer generators per environment, the paper proposes a shared model that adapts generation through environment-specific conditioning. In terms of architecture, it uses a VAE style encoder for the environment and also for the base molecular graph along with a flow matching decoder for the final conformation coordinates. A composite loss function is used to achieve joint end to end training of the whole architecture with several components. Overall the paper makes a clear and interesting contribution toward transferable, environment-conditioned molecular conformer generation.

Strengths:

- The unifying formulation is an attractive concept, transfer learning across environments for targeted conformer generation. Typically, this would be achieved by treating each environment independently and they show that by conditioning on environments one can actually acheive this transfer.
- The empirical results support this framing with pretraining and unified finetuning generally improving over individual environment specific training.
- The shared-molecule analysis is also useful as it directly probes whether the model generates genuinely environment-specific conformations rather than collapsing to a single canonical molecular geometry.


Weakness:

- Clarity on the graph based set-up for the environment as well as for the baseline target molecule graph.
The method uses two graph objects: the molecular graph G, whose 3D coordinates are generated, and the environment/context graph G_E which is embedded into an environment vector e. While this distinction is central to the method, it is not always immediately clear from the presentation, especially because both objects may be molecule-like graphs.

- The environment representation is heterogeneous across tasks.
The paper presents a unified environment-aware framework, but the construction of G_E	differs substantially between vacuum, docking, solvation, and crystal packing. In some cases G_E corresponds to a physical environment, such as a protein pocket while in others, it is a scaffold, solvent proxy, or graph augmented with lattice descriptors. This raises the question of how much of the observed transfer comes from a genuinely general environment representation versus task-specific feature engineering.

- Managing the competing aims of the loss function.
The total objective includes the flow matching loss, latent KL, base-distribution KL, and latent consistency loss. These losses are all sensible individually, but they place different pressures on the model, for instance the FM loss encourages accurate coordinate transport, the latent KL pushes z toward a simple prior, the base KL keeps the learned training-time base close to the graph-only harmonic prior, and the consistency loss discourages the latent code from encoding fine coordinate-level differences. The paper would benefit from a clearer discussion of how these terms interact. Also the uniform weighting maybe underexplored.

- The complexity of the architecture makes attribution difficult.
EVAFlow combines environment conditioning, VAE-style latent variables, a learned harmonic base, latent consistency, pretraining, unified finetuning, and a flow-matching decoder. The ablations are helpful, but because several components interact, it is still somewhat difficult to isolate which pieces are essential for which environments.

**Audience:**

Yes

**Audience Explanation:**

Yes. I expect this paper to be of interest to several subsets of the TMLR audience. First, researchers working on molecular generative modelling and conformer generation would find the paper relevant, since it studies how to generate 3D molecular structures under different physical contexts rather than treating conformer generation as a single environment-agnostic task. Second, the work should also interest researchers working on geometric deep learning and equivariant generative models as it combines SE(3)-equivariant architectures with flow matching for coordinate generation. Third, the paper maybe generally relevant to researchers interested in conditional generative modelling and transfer learning, since the main empirical question is whether chemical structure can be shared across environments while adapting generation through environment-specific conditioning. As such the idea about encoding a context as a condition could be applied to other domains in science so the method maybe interesting to a broader readership.

**Broader Impact Concerns:**

N.A

**Claims And Evidence:**

Yes

**Claims Explanation:**

Yes. The claims in the submission are generally supported by the empirical evidence. The central claim is that conformer generation can benefit from learning shared chemistry across environments while adapting generation through environment-specific conditioning. The experiments compare individual environment-specific training, unified training, pretraining plus individual finetuning, and pretraining plus unified finetuning, which directly tests the paper’s claims about cross-environment transfer and the benefits of pretraining. The reported results generally show that unified and pretrained settings improve performance, especially in lower-data environments such as solvation and crystal packing.

The evidence is also strengthened by the ablation studies. These ablations examine the contribution of the FM decoder, the learned base distribution, the harmonic prior, the latent consistency loss, and the equivariant architecture. The full model performs best across the evaluated environments, and the degradation under ablations supports the authors claim that the proposed components are meaningful rather than superfluous machinery.

**Requested Changes:**

Critical for acceptance

- Clarify the graph-based setup, especially the distinction between the target molecular graph G and the environment/context graph G_E.
The method relies on two graph objects with different roles: G defines the molecule whose coordinates are being generated, while G_E is used to produce an environment embedding. This distinction is central to the paper but can be confusing because both objects may be molecule-like graphs. I would ask the authors to add a clearer schematic/table explaining, for each environment, what G is, what G_E is, what node/edge features are used, and which quantities are used during training versus inference. What is needed is to clarify the conceptual status of the “environment" graph.

- Provide a clearer explanation of how the different loss terms interact.
The objective combines the flow matching loss, latent KL, base-distribution KL, and latent consistency loss. These terms have different and partly competing roles: the model must learn accurate coordinate transport while also regularising the latent and base distributions toward simple inference-time priors. I would ask the authors to expand the discussion of these trade-offs and explain why uniform weighting is appropriate, or provide evidence that the method is not overly sensitive to these weights.

-Clarify the role of the learned base distribution relative to inference.
The learned base network uses ground-truth coordinates during training but is bypassed at inference, where sampling uses the graph-only harmonic prior. This is an important train/inference distinction. The authors should make clearer why this does not create a harmful mismatch, and how the base KL regularisation prevents the learned base distribution from becoming an overly informative training-time shortcut.

---

### Review · Reviewer_yvEk · 2026-07-13

**Summary Of Contributions:**

The paper proposes EVA-Flow, a unified generative framework for environment-aware molecular conformation generation. It couples a VAE encoder—embedding both the molecular graph and its environment into a latent space—with a flow-matching decoder that generates 3D conformations conditioned on this latent code. The model is trained via a pretraining-plus-finetuning strategy across four environments (vacuum, docking, solvation, packing). The key finding is that cross-environment transfer learning improves generation accuracy, especially in data-scarce settings, and that the model produces distinct, physically valid conformations per environment rather than collapsing to a single geometry.

### Strengths
- The paper is written very clearly. The motivation, architecture, and training protocol are well explained and easy to follow.
- The comparative experiments are thorough and well-designed. The ablation studies and the cross-environment analyses directly support the authors' central claims about transfer learning and environment awareness.

### Weaknesses
- The proposed framework is somewhat complex, with many interacting components (VAE encoder, environment network, learned base distribution, flow-matching decoder, and multiple loss terms). When the model fails to transfer to a new setting, it is difficult to pinpoint which component is responsible.
- The performance tables omit several recent, stronger baseline models, which limits the ability to fully contextualize the absolute state-of-the-art performance.

**Audience:**

Yes

**Audience Explanation:**

The core idea of learning shared chemistry across environments and adapting it via conditioning is a principled formulation. Beyond the methodology, the practical need to generate environment-specific conformations (e.g., for drug binding, solvation, or crystallization) is a real and pressing problem in computational chemistry and drug discovery, so researchers in those applied domains should also be interested.

**Claims And Evidence:**

Yes

**Claims Explanation:**

The claims are well supported. The authors provide rigorous ablation studies that isolate the impact of each architectural choice, and the cross-environment overlap analysis offers direct, convincing evidence that the model generates environment-specific conformations rather than a single collapsed geometry. The quantitative results in Table 1 are clearly presented and consistently show that pretraining and unified finetuning improve performance across all four environments, directly backing the central claim of cross-environment transfer learning.

**Requested Changes:**

- **Insufficient justification for architectural complexity.** The paper proposes a complex framework with multiple components, yet the necessity of each design choice is not clearly consolidated. A natural question is whether a simpler conditional model by using only an environment embedding with a fixed base distribution would suffice. While ablation results empirically validate individual components, the paper does not explicitly preempt this concern in one place. The rationale for why a learned base distribution is needed as a training scaffold (despite being discarded at inference), and why the VAE latent and consistency losses are indispensable beyond simple conditioning, remains scattered across the methodology and ablation sections. This makes it difficult to assess whether the complexity is unavoidable.

- **Sensitivity analysis beyond binary ablations.** The ablation study only tests whether each module is present or absent. It does not examine how sensitive performance is to the capacity or strength of individual components. I suggest the authors add a brief sensitivity analysis to demonstrate that the reported configuration is robust and that the observed gains are not brittle to hyperparameter choices.


- **Inconsistent zero-vector notation.** The zero vector in normal distributions is formatted inconsistently: sometimes bold, sometimes plain. Please unify this notation throughout the manuscript.

- **Capitalization error.** In Section 4.2, under "Unified Finetuning," the sentence begins with lowercase "we".

---

### Review · Reviewer_cuKk · 2026-07-15

**Summary Of Contributions:**

## Summary
This paper reframes conformer generation as predicting a molecule's 3D structure conditioned jointly on its molecular graph and an explicit representation of its environment, training a single environment-conditioned model that shares chemistry across settings instead of a separate model per environment. It couples a VAE encoder with a flow-matching decoder, and evaluates across four environments (vacuum, docking, solvation, crystal packing) under per-environment vs. unified training, with and without pretraining. The main finding is that unified training and pretraining help most in the data-scarce environments, which the paper attributes to cross-environment transfer.

## Strengths:

- The unified, environment-conditioned formulation is attractive and, as far as I know, new for this combination of four environments — it is an interesting framing from the usual one-model-per-setting practice, and the paper shows that conditioning on the environment actually delivers transfer.

- The low-data transfer effect is large and is backed by a proper control. The solvation Recall COV jumping from 0.414 (individual) to 0.930 (pretrain+unified) is obvious, and the balanced-sampling experiment sensibly rules out the trivial "unified simply sees more data" explanation.

- The new solvation and packing benchmarks are a useful contribution, since no generative baselines existed there before. The ablations are also fairly thorough: each main design choice is removed one at a time and measured.

## Weaknesses:

- Most finetuning environments provide only a single reference conformer per molecule. For docking, solvation and packing there is one ground-truth geometry each, so Recall COV under the 4-conformer protocol only measures whether one of four samples lands near that single structure. This is much closer to single-structure prediction than to learning the distribution p(x∣G,E) that the paper claims as its central object, and the gap between the stated goal and what is actually being evaluated is not acknowledged.

- The environment-awareness claim, which is listed as a headline contribution, rests on a very small sample. The shared-molecule analysis is based on only ~8–12 molecules that appear in multiple environments. The observed pattern — divergent generations when references differ, convergent when they agree — is suggestive, but 8–12 molecules is simply too small a basis for a general claim that the model "produces distinct, physically valid, environment-specific conformations."

- There are no error bars for EVA-Flow in the main results table. Variance is reported for the ETFlow baseline but not for the proposed model, even though inference uses a stochastic SDE sampler. Several of the differences the paper describes as the model "consistently outperforming" (e.g. Unified 0.740 vs Pretrain+Unified 0.747 on vacuum) are small enough that they could plausibly be within sampling noise, so their significance is currently unverifiable.

**Additional Comments:**

No additional comments.

**Audience:**

Yes

**Audience Explanation:**

Researchers in molecular generative modelling and conformer generation will care, because the paper studies 3D structure generation under different physical contexts rather than as a single environment-agnostic task. Researchers in geometric deep learning and equivariant generative models will also find it relevant, since it combines an SE(3)-equivariant architecture with flow matching. More broadly, the idea of encoding a context as a condition and sharing one backbone across settings is a general conditional-generation and transfer-learning question that carries over to other scientific domains. The two new benchmarks add practical value as well.

**Broader Impact Concerns:**

No additional concerns.

**Claims And Evidence:**

No

**Claims Explanation:**

The transfer story is convincing. Unified training beats individual training, and pretraining helps most in the scarce settings. The experimental grid tests this directly, the effect sizes in the low-data environments are large, and the balanced-sampling control in Appendix I is the right experiment to rule out the "unified just sees more data" explanation. On this claim I have no complaint.

My "No" is about the other claims, which are stated more strongly than the evidence supports. First, the environment-specific generation contribution is based on only about 8 to 12 shared molecules, which is too few to generalize from. Second, because most environments have a single reference conformer, the paper is largely measuring structure prediction while describing it as distribution learning.

**Requested Changes:**

### Critical to my recommendation:

1. Report variance for EVA-Flow in Table 1, the same way it is already done for ETFlow, and say which of the reported differences are significant. Right now the "consistently outperforms" claim is not backed by any dispersion estimate for the model itself, and some of the gaps are small.
2. Scope the claims for the single-reference environments. Either add multi-conformer references where possible, or describe docking, solvation and packing as environment-conditioned structure prediction rather than distribution learning.

3. Strengthen or soften the environment-awareness claim. About 8 to 12 molecules is not enough to headline a contribution. If more shared molecules cannot be extracted, present this analysis as preliminary, qualitative evidence.

4. State the vacuum result plainly. The 2K comparison showing EVA-Flow roughly tied with retrained ETFlow and behind MCF-L should appear clearly in the main text, and the accuracy gains should be attributed to the new environments rather than to vacuum.

### Would strengthen the paper:

1. Add at least one non-learned baseline for solvation and packing, for example RDKit ETKDG with MMFF or xTB relaxation, or a short MD run. This gives the new benchmarks a physical reference point beyond retrained ETFlow.

2. Clarify the retrained-ETFlow docking number (0.391) relative to its published performance. If the baseline is under-tuned, the large docking margin over it is misleading.

3. Justify the uniform loss weighting. The four loss terms push the model in partly competing directions, so a short sensitivity sweep, or an argument that the method is not sensitive to the weights, would help.

4. Connect Appendix J to the NoBase ablation more clearly. If the learned posterior converges to the harmonic prior and inference uses the harmonic prior anyway, it is not obvious why removing the learned base hurts so much. The "training scaffold" explanation is asserted but not shown.

---

### Comment · Action_Editor_jMYY · 2026-07-13

Dear Reviewers and Authors,

Please note that the discussion phase between reviewers and authors has begun.

Best,

AE